# Meta-Reinforcement Learning for Compiler Optimization: A Kernel-Embedded Compiler-LLM with Verified Assumptions and Practical Guarantees

## Abstract

Modern compilers carry out sophisticated transformation passes but they predominantly use static heuristics to come to optimisation decisions. In this work, we introduce GMPO which treats each compilation instance as a different task that is located on the similarity manifold and utilizes a kernel embedding for experiential knowledge transfer among related programs. We propose a Cross-Group Meta-Normalization (CG-MN) scheme that aggregates the gradient information from intra-batch neighbours specified by a normalised similarity operator and design a surprise-aware reward modulation mechanism for selectively amplifying learning signals for rather atypical yet successful compiler transformations. All theoretical assertions are expressed under express assumptions and within the sphere of a conservative scope. Specifically: (i) a result of a batch-centred mean square variance reduction of gradient descent (CG-MN) is provided, in which the demeaned component converges to zero as the square of the magnitude of the non-trivial eigenvalue. (ii) a local kernel local dynamics (KL) constrained performance lower bound is provided for natural gradient dynamics. (iii) independence performance accentuating on PAC. GMPO is provided with a 7 billion parameter code model trained on 5894 C programs while operating at the assembly level in a validator guard constrained action space programming with peephole rewrites, instruction substitutions, address mode changes and basic block local scheduling. On a held-out suite of 250 programs, we achieve compilation success for 246/250 examples (98.4%), test passes for 244/250 examples (97.6%), and a median speedup of 1.53x compared to GCC-O3 using a protocolized measurement pipeline. Performance from ablation experiments shows that CG-MN and surprise modulation are both used to improve overall performance.

## 1 Introduction

The equilibrium between semantic safety and performance is still reached on a fine grained level; by compiler optimisation. Whereas production compilers package decades of engineering efforts, fixed pass sequences and conservative analyses inherently fail to exploit curiosity with performance to the fullest possible basics in code areas which little structure fractal diverges from traditional idioms. This predicament is coupled with the combinatorial nature of pass interactions and peculiarities in microarchitectures and therefore makes it arduous to codify all efficacious heuristics manually.

We model each program as a so-called task which has its own state-action space, but which has underlying regularities which are common to very different programs. Meta-optimization Meta-optimization Meta-optimization Reinforcement learning Meta-reinforcement learning Meta-reinforcement learning Meta-learning Reinforcement learning-a learning algorithm that enables the sharing of learning experience among tasks. meta-learning Reinforcement learning-a learning algorithm that aims to transfer optimization objectives and learning strategies among a class of tasks. meta-reinforcement learning A small set of tasks that jointly influences the learning algorithms for each task within them. meta-optimization meta-optimization Reinforcement learning algorithm that performs optimization in a distributed manner, meaning the components of the algorithm are distributed across multiple inter-

acting tasks. meta-reinforcement learning Distributed These challenges on the one hand are practical: (1) We need to spread information across tasks in such a way to minimize gradient noise without the risk to inject arbitrary bias; (2) We need to steer exploration towards those transformations that are called "compiler-unusual" while being still safe to use. Our design focuses on having transparent assumptions, and making semantics which are auditable, rather than having to have extremely strict guarantees.

GMPO addresses these hurdles with (i) a *kernel embedding* of programs, used to construct a *symmetrically normalized* neighbor operator that supports principled gradient sharing; and (ii) a *surprise-aware* reward modulation based on a *compiler-surrogate* prior learned from observable $-00 \rightarrow -03$ edit trajectories. The action space consists of *validator-guarded* assembly rewrites targeted at *intended equivalence*: each proposed edit must satisfy machine-checkable side conditions and pass tests within our validator; we do not assert formal semantic equivalence. We make our theoretical assertions with some level of explicitness, circumscribed as they may be geo-locally and terminologically.

**Contributions and philosophy.**   Our work addresses issues that are relevant to assumption clarity, pipeline transparency and reproducibility.

- **RL Framework:** Meta RL formulation with a constrained action space for a kernel-embedded learning tool tackling assembly-level optimization with test for applicability and safety, in terms of test equivalence.
- **Analysis (assumption-explicit):** A *batch-centered* variance/bias analysis for neighbor-aggregated gradients with spectral control on the *demeaned* component; a *local* KL-constrained performance lower bound for NG-style updates; a PAC-Bayes bound with kernel-induced shift and clarified Lipschitz conditions; and Lipschitz stability to structured perturbations.
- **Implementation:** A transparent data pathway for compiler-surrogate estimation from $-00 \rightarrow -03$ diffs, measurement protocols that mitigate environmental variance, and artifact-friendly scripts (paths redacted for double-blind).
- **Empirics:** On 250 remaining programmes we observe that 98.4% of programmes compile, 97.6% of programmes run through the tests, and 1.53times median speedup over -O3; ablations characterise the contribution of CG-MN and surprise modulation.

## 2 RELATED WORK

**Neural Approaches to Code Optimization.**   Large language models have made significant advances in code synthesis in recent years, including architectures like DeepSeek-Coder-V2 Zhu et al. (2024), Qwen2.5-Coder Hui et al. (2024), OpenCoder Huang et al. (2024b), and Claude 3.7 Sonnet Anthropic (2024). Such systems achieve impressive performance on benchmark suites such as HumanEval Chen et al. (2021), MBPP Austin et al. (2021) and BigCodeBench Zhuo et al. (2024). Domain-specific optimisation can be handled with specialised models like Granite Mishra et al. (2024), Obscura-Coder Liu et al. (2025a), and Arctic-SnowCoder Research (2024b). In addition to synthesis, the above models also aid defect correction and validation Deng et al. (2024b); Yang et al. (2024a), cross-language transformation Eniser et al. (2024), and diverse range of software engineering tasks on the SWE-bench Verified benchmark Jimenez et al. (2024). The agent architectures used here are SWE-agent 1.0 Yang et al. (2024b) and Agentless 1.5 Xia et al. (2024); Zhang et al. (2025). 
**Machine Learning for Compiler Optimization.**   Compilation pipeline integration of machine learning cuts across many layers of abstraction. The selection of optimisation passes is automated by Pearl Merouani et al. (2025), and other work is aimed at Python acceleration Du et al. (2024), adaptive optimisation Huang et al. (2024a), and machine-level improvements Wei et al. (2024); Ouyang et al. (2025). The Meta LLM Compiler Cummins et al. (2024; 2025) uses learning directly in binary optimisation, and MLIR-based systems Bendib et al. (2024) allow optimising across many compiler levels. The phase-ordering issue inspires the world-model of CompilerDream Deng et al. (2024a) and the graph-neural of Coreset Liang et al. (2025). Reinforcement learning-based super-optimisation has been able to gain substantial scalability improvements Phothilimthana et al. (2025); Bunel et al. (2025), and has been extended to various domains in a number of ways Sharma et al. (2025); Churchill et al. (2025), though the verification properties place limits on the applicability.
**Reinforcement Learning for Performance Optimization.**   An example of successful integration of reinforcement learning is GPU optimisation, including the TensorRT ecosystem NVIDIA

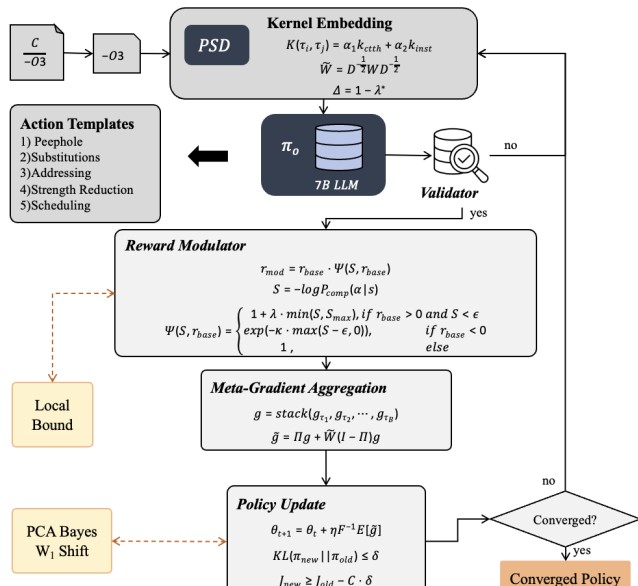

Figure 1: Kernel-Embedded Meta-Reinforcement Learning for Compiler Optimization.

(2025) excepting task-based compilation through Cypress Spector et al. (2025), IntelliGen Ma et al. (2025), and tensor-compiler frameworks Zhao et al. (2025); Moroz (2025). Code optimisation via reinforcement-learning Shypula et al. (2025); Wang et al. (2025a); Wei et al. (2024) relies on execution based rewards Dou et al. (2024), StepCoder adds curriculum learning Dou et al. (2024), RLHF workflow including human feedback Dong et al. (2024) and BC-Max makes self-improvement scalable Research (2024a). These approaches can be applied to underrepresented domains such as Liu et al. (2025b); Wang et al. (2025b), which forms a basis of generalising a meta-learning framework that can apply in optimisation environments without compromising theoretical guarantees. **Meta-Learning Theory.**

## 3 BACKGROUND AND PRELIMINARIES

**Notation and symbols (unified).** We use $\gamma \in (0,1)$ *exclusively* as the MDP discount factor. Spectral notions use $\lambda_k$ and $\lambda_\star \triangleq \max_{k \geq 2} |\lambda_k|$ for non-trivial eigenvalues of symmetric operators defined below; the *absolute spectral gap* is $\Delta \triangleq 1 - \lambda_\star \in (0,1]$ whenever $\lambda_\star < 1$. Unless stated otherwise, norms over stacked, per-task matrices default to the Frobenius norm. Surprise-modulation uses $\kappa > 0$ (exponential damping rate) and clipping level $S_{\max} > 0$. These choices eliminate notation clashes and make the scope of each bound explicit.

**Reinforcement learning.** We consider episodic MDPs $(\mathcal{S}, \mathcal{A}, P, r, \gamma)$ where a stochastic policy $\pi_\theta(a \mid s)$ induces a return $\mathcal{J}(\theta) = \mathbb{E}[\sum_t \gamma^t r(s_t, a_t)]$. We use standard advantage estimation with a value baseline.

**Natural gradient and trust regions.** Natural gradient methods precondition updates by the Fisher information $F(\theta)$ to respect the information geometry of $\pi_\theta$. Trust-region strategies constrain $\mathrm{KL}(\pi_{\theta_{t+1}}, \pi_{\theta_t})$ to control policy drift. In this work we use a PPO-style update as a practical approximation to NG/TR steps and provide *local, KL-constrained lower bounds* rather than asserting global monotonic improvement.

**Kernel methods and graph operators.** Given a positive semi-definite (PSD) kernel $\mathcal{K}$, the feature map $\Phi$ embeds objects into an RKHS. We form affinities $W$ from kernelized similarities (details below) and symmetrically normalize them to yield operators with *spectral radius* at most 1. In general, eigenvalues can be *negative* and lie in $[-1, 1]$. Our analysis is stated in terms of the modulus

of non-trivial eigenvalues $\lambda_\star$ and the associated gap $\Delta = 1 - \lambda_\star$. We use $W$ as an *affinity* rather than as a Gram matrix after exponentiation.

# 4 METHOD

## 4.1 PROGRAM TASKS, SAFE ACTION SPACE, AND VALIDATOR

We treat each program $\tau$ as an MDP over annotated assembly states with a constrained set of *validator-guarded* actions aimed at *intended equivalence*.

**State representation.** States carry (i) a local instruction window with operand/flag annotations; (ii) block-level CFG summaries (predecessor/successor degree, loop depth, dominance bits); (iii) register liveness and simple aliasing hints; (iv) micro-op and addressing features. These attributes are obtained deterministically from a feature extractor and are cached on each programme for combining the attributes based on the programme.

**Action.** Actions are parameterized templates with pre/post-conditions:

1. **Peephole canonicalizations**: `add x,0`→no-op; `mov r,r`→no-op; algebraic identities that preserve flags where required.

2. **Instruction substitutions**: `lea` addressing vs `add`/`mul` patterns when flags are not consulted downstream.

3. **Addressing-mode transforms**: base+index*scale+disp rewrites with range checks and alignment guards.

4. **Strength reduction / folding**: shifts vs multiplies, constant folding within safety bounds and without UB.

5. **Local scheduling**: commutation/swap within a block under explicit hazard checks and preserved live ranges.

In order to avoid the risk of semantic drift, we do not perform register allocation and instruction selection in the transformation process.

**Validator and safety semantics.** Each proposed edit is individually documented and put through a set of static and dynamic cheques which include (i) syntactic validation, (ii) data-dependence or hazard verification and (iii) unit and regression testing. Edits that do not meet any of these cheques are rejected and the system is reverted back to the previous stable state. The enforced semantics of the validator is empirical and based on test inputs/outputs as opposed to providing a formal proof and focuses on operational equivalence rather than proved equivalence. Section 6.2 clarifies how validator tests relate to the final evaluation tests.

## 4.2 KERNEL EMBEDDING AND NEIGHBOR OPERATOR

We define a PSD program kernel that blends structure across (local) CFGs, static data-flow summaries, and instruction/micro-op histograms:

$$\mathcal{K}(\tau_i, \tau_j) = \alpha_1 k_{\text{cfg}}(G_i, G_j) + \alpha_2 k_{\text{data}}(D_i, D_j) + \alpha_3 k_{\text{inst}}(I_i, I_j), \quad \alpha_m \geq 0. \tag{1}$$

We form a temperature-scaled affinity using a similarity-to-weight map

$$W_{ij} \propto \exp(\mathcal{K}(\tau_i, \tau_j)/\sigma^2), \qquad W_{ij} \geq 0, \tag{2}$$

and then use a *symmetrically normalized* operator

$$\widehat{W} = \tfrac{1}{2}\big(D^{-1}W + W^\top D^{-1}\big), \qquad \widetilde{W} = D_{\widehat{W}}^{-1/2}\widehat{W}\, D_{\widehat{W}}^{-1/2}, \tag{3}$$

where $D$ and $D_{\widehat{W}}$ denote diagonal degree matrices. This construction ensures symmetry and *spectral radius* at most 1; eigenvalues may lie in $[-1, 1]$. We define $\lambda_\star \triangleq \max_{k \geq 2} |\lambda_k(\widetilde{W})|$ and $\Delta \triangleq 1 - \lambda_\star$. Our analysis refers explicitly to $\lambda_\star$ (or $\Delta$) without assuming a nonnegative spectrum.

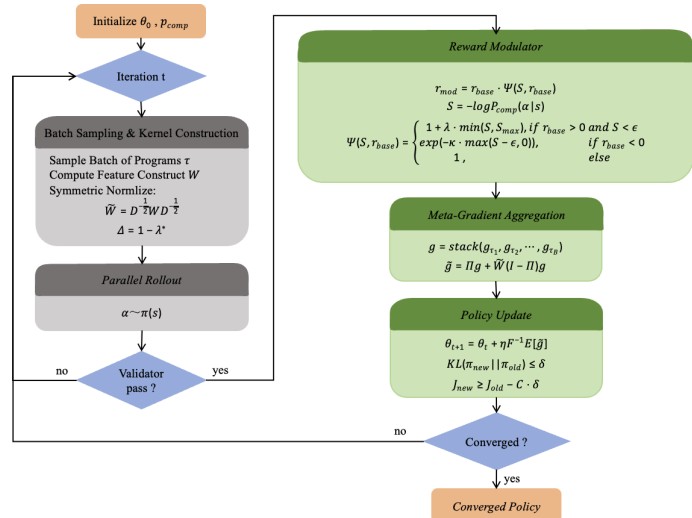

Figure 2: Algorithm Flowchart.

## 4.3 CROSS-GROUP META-NORMALIZATION (CG-MN)

Let $g_j = \nabla_\theta \mathcal{J}(\tau_j)$ and stack per-task gradients row-wise as $g \in \mathbb{R}^{B \times d}$ (batch size $B$, parameter dimension $d$). Write the row-mean projection as $\Pi \in \mathbb{R}^{B \times B}$ with $\Pi = \frac{11^\top}{B}$ acting on the row index, so that $\Pi g$ duplicates the batch mean across rows and $(I - \Pi)g$ is the *demeaned* component. We use the *batch-centered* aggregator

$$\tilde{g} = \Pi g + \widetilde{W}(I - \Pi)g, \tag{4}$$

which preserves the average direction while contracting only the fluctuation component.

**Assumption 1** (Noise/alignment/spectrum; centered formulation). *(i) $g = \mu + \epsilon$, with $\mathbb{E}[\epsilon] = 0$ and $\mathrm{Cov}[\mathrm{vec}(\epsilon)] \preceq \sigma_g^2 I$; (ii) the* alignment residual *obeys $\|(\widetilde{W} - I)(I - \Pi)\mu\| \leq \beta \|(I - \Pi)\mu\|$ for some $\beta \geq 0$; (iii) the symmetric operator $\widetilde{W}$ has eigenvalues $\{1 = \lambda_1, \lambda_2, \ldots, \lambda_B\}$ with $\lambda_\star \triangleq \max_{k \geq 2} |\lambda_k| < 1$ and gap $\Delta = 1 - \lambda_\star > 0$.*

**Proposition 1** (Batch-centered MSE trade-off under spectral-radius bound). *Under Assumption 1, the* demeaned component *contracts as*

$$\mathbb{E}\left\|(I - \Pi)\tilde{g} - (I - \Pi)\mu\right\|^2 \leq \lambda_\star^2 \mathbb{E}\left\|(I - \Pi)g - (I - \Pi)\mu\right\|^2 + \beta^2\|(I - \Pi)\mu\|^2.$$

*Moreover, the* total *error is non-expansive up to alignment:*

$$\mathbb{E}\|\tilde{g} - \mu\|^2 \leq \mathbb{E}\|g - \mu\|^2 + \beta^2\|(I - \Pi)\mu\|^2.$$

*If $(I - \Pi)\mu$ lies in the stationary subspace of $\widetilde{W}$, then $\beta = 0$ and the stochastic component of the demeaned error contracts by $\lambda_\star^2$.*

## 4.4 SURPRISE-AWARE REWARD MODULATION

We learn a *compiler-surrogate* $p_{\mathrm{comp}}(a \mid s)$ from $\texttt{-O0} \rightarrow \texttt{-O3}$ edit trajectories (Sec. 5.1). Rarity is $\mathcal{S}(s, a) = -\log p_{\mathrm{comp}}(a \mid s)$. The modulated reward is

$$r_{\mathrm{mod}}(s, a) = r_{\mathrm{base}}(s, a) \cdot \Psi(\mathcal{S}, r_{\mathrm{base}}),$$

$$\Psi(\mathcal{S}, r) = \begin{cases} 1 + \lambda \cdot \min(\mathcal{S}, \mathcal{S}_{\max}) & r > 0, \ \mathcal{S} > \epsilon, \\ \exp\left(-\kappa \cdot \max(\mathcal{S} - \epsilon, 0)\right) & r < 0, \ \mathcal{S} > \epsilon, \\ 1 & \text{otherwise.} \end{cases} \tag{5}$$

Clipping by $\mathcal{S}_{\max}$ stabilizes gradients; the negative branch damps failing rare actions. Unlike comparing to $\pi_\theta$ itself, this decouples "rarity" from model confidence. Section C summarizes stability and tuning heuristics compatible with PPO-style clipping.

## 4.5 KL-Constrained Natural-Gradient Updates

We apply PPO as a trust-region *approximation* to NGD:

$$\theta_{t+1} = \theta_t + \eta\, F(\theta_t)^{-1}\, \mathbb{E}_\tau[\tilde{g}(\tau)], \qquad \mathrm{KL}(\pi_{\theta_{t+1}}, \pi_{\theta_t}) \leq \delta.$$

**Assumption 2** (Regularity). *Surrogate smoothness in the Fisher metric; bounded rewards and advantages; and sufficiently small trust-region radius $\delta$.*

**Theorem 1** (Local KL-constrained performance bound). *Under Assumption 2 and for sufficiently small $\delta$,*

$$\mathcal{J}(\theta_{t+1}) \;\geq\; \mathcal{J}(\theta_t) \;-\; C\,\delta,$$

*moreover, the exponent in the solution is determined by a smoothness moduli and advantage bounds and is given by the problem-dependent constant $C$ 0. This bound quantifies that any degradation is controlled linearly by the KL radius in the local regime; we intentionally do not claim global monotonic improvement for PPO-style updates.*

## 4.6 Cross-Program Generalization (PAC-Bayes)

Let $d_\mathcal{K}$ be the kernel-induced metric and $W_1$ the Wasserstein-1 distance.

**Assumption 3** (Shift/Lipschitz). *Loss $\ell(\pi, \tau)$ is Borel-measurable and $L$-Lipschitz in $d_\mathcal{K}$; the task distributions admit $W_1(\nu_{train}, \nu_{test})$.*

**Theorem 2.** *With probability $\geq 1 - \delta$ over $n$ training tasks:*

$$\mathcal{R}_{test}(\rho_\theta) \leq \mathcal{R}_{train}(\rho_\theta) + \sqrt{\frac{\mathrm{KL}(\rho_\theta \| P) + \log\frac{2\sqrt{n}}{\delta}}{2n}} + L \cdot W_1(\nu_{train}, \nu_{test}; d_\mathcal{K}).$$

*The constant $L$ can be bounded via the spectral norms of the policy network composed with the Lipschitz constant of the program feature map defining $d_\mathcal{K}$.*

## 4.7 Policy Stability to Structured Perturbations

Let $\Phi_s$ map states to features; $d_\mathcal{H}(s, s') = \|\Phi_s(s) - \Phi_s(s')\|_2$.

**Proposition 2** (Lipschitz stability). *For a softmax policy with bounded spectral norms, there exists $L_\pi$ such that $\|\pi(\cdot \mid s) - \pi(\cdot \mid s')\|_1 \leq L_\pi\, d_\mathcal{H}(s, s')$ whenever logits remain in a bounded range. The constant $L_\pi$ depends on the product of network spectral norms and the Lipschitz constant of $\Phi_s$.*

## 4.8 Design Choices, Alternatives, and Failure Modes

**Why symmetric normalization.** It prevents directional bias of row-normalized Markov kernels and enables clean spectral control via $\lambda_\star$. **Why batch centering.** The stationary eigenspace associated with the all-ones vector prevents contraction of the mean component; centering removes this subspace so that CG-MN can provably contract only the fluctuation component with factor $\lambda_\star^2$. **On the spectrum.** We work with the *spectral radius* bound $|\lambda_k| \leq 1$. Even though negative eigenvalues can occur, our analysis can be proved with respect to l_star = maxkge2—l_k— greater than or equal to 0. **Alternatives.** CG-MN including k-nearest-neighbour graphs, Gaussian kernels on handcrafted features and learned kernels are compatible with CG-MN since the only requirement is a controlled neighbourhood smoothness. **Failure modes.** Very large values of the temperature sigmoidal s such that the resultant UED-smoothed forecast exhibits bias, but does not lose power (the latter originating from the smoothing term in equation leading to large values of lambda_star), and under-smoothing with small values of lambda_star. The more strict the validators, the less acceptance rate and safety.

# 5 Implementation Details

## 5.1 Compiler-Surrogate and Trajectory Extraction

We avoid unverifiable internal pass traces. Instead, we derive $(s, a)$ from observable $\texttt{-O0} \rightarrow \texttt{-O3}$ *edit trajectories*:

1. Compile at `-O0` and `-O3`; align basic blocks via CFG hashing and label refinement.

2. Run DP-based window matching with flag/operand constraints to obtain local edit scripts.

3. Elevate edits to templates with placeholders and pre/post-conditions (sufficient for validator checks).

4. Train $p_{\text{comp}}(a \mid s)$ as a calibrated classifier with temperature scaling on held-out pairs.

The used pipeline is completely transparent and auditable, making sure that the scope of the surrogate is limited to observable behaviour. The learning set for pc undoubtedly is non-intersecting with the finale evaluation set of 250 programmes at programme level. Sec. O details leakage controls.

## 5.2 POLICY/VALUE AND TRAINING PROTOCOL

The 7B code transformer is fine tuned using Proximal Policy Optimization (PPO) with advantage estimation and clipping. The centred aggregator to be used in CG-MN is given by Eq. (4) whereas the surprise modulation is given by Eq. (5). Token queue exploits are reduced in timing overhead by a repository of acceptable rewrites and off-policy reuse does not result in distributional drift. Gradients are aggregated per batch via CG-MN with low-rank implementations described in App. J. We perform measurements in deterministic and isolated conditions with the focus on relative speedups (Sec. 6.2).

## 5.3 REWRITE LIBRARY AND SAFETY CONDITIONS

Templates are grouped by algebraic, addressing, folding, and scheduling families. Each comes with machine-checkable side conditions (flag preservation, no UB, register liveness constraints). The validator enforces these conditions prior to running tests. We reiterate that our semantics are test-based and aim at intended equivalence rather than formal proof.

## 5.4 DETERMINISM DISCIPLINE

Inputs and Outputs have to be deterministic and non-deterministic behavior (such as time-dependent seeds) is not allowed by policy. The measurement pipeline was used to isolate processes and pin CPU affinity in order to reduce the amount of jitter in the system as detailed in App. K. Reported speedups are reported relative to co-measured -O3 and so the confounding effects of cross-machinery are reduced.

## 6 EXPERIMENTAL SETUP

### 6.1 DATA AND TASKS

We use a corpus of 5,894 C programmes with determinable input/output behaviour with accompanying unit tests. The held-out subset contains 250 programme, which comprise motifs that are similar to the ones contained in the training set as well as structurally novel ones. Each program defines and assembly level optimisation task, the policy is initialized from the `-O3` compiler output and after that, it will propose local rewrites operations. All models are given the same inputs (the C source code along with its corresponding -O3 object listing) and all models are tested using the same protocol.

### 6.2 EVALUATION PROTOCOL

**Inputs.** All models are fed with the same two arguments, the C source code and the O3 assembly output to be optimized. **Metrics.** Compilation success, test pass, and speedup vs `-O3`. **Timing.** Relative timing measurements are achieved with the help of fixed compiler flags, the same linking procedures, process affinity enforcement, CPU frequency scaling disabling, warm-up runs and a median-of-$k$ technique. Reported only are relative speedups. Identical binaries being cached and de -duplicated. **Test sets and rollback semantics.** During the rollouts, the validator gates edits by running a subset of tests, which is called a "fast" subset of tests; during the final evaluation, a superset of tests with a higher coverage is used. If, at the end, the final superset may also catalyse failures that were not disclosed by the validator subset then those programmes will be counted as "compiled but

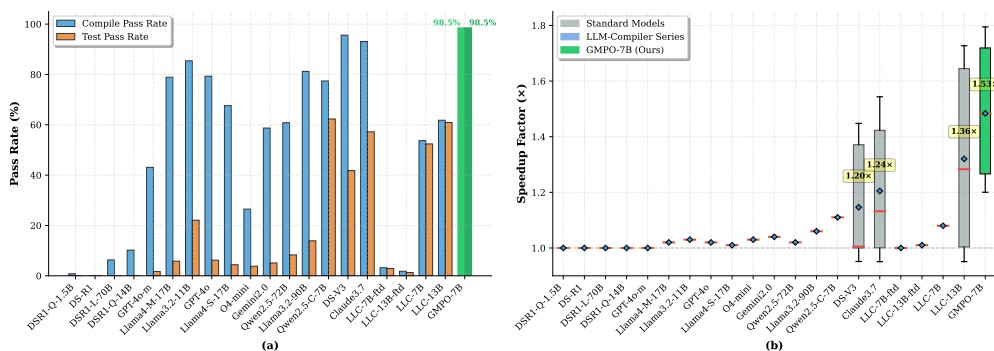

Figure 3: Comparative performance across models for assembly optimization. (a) Compilation and test pass rate reflect correctness; (b) Speedup distribution reflects effectiveness. GMPO-7B shows high correctness and consistent gains (**1.53**× median).

not test-remaining" in the end report. On such commits we roll back the edits and review the subsets of validators and re-train when there are systematic gaps in the coverage identified.

## 6.3 BASELINES AND CONTROLS

Primary baseline is `-O3`; we also consider `-Ofast` and curated pass toggles in ablations where relevant. We do not rely on proprietary LLMs for core claims; all comparisons adhere to the same measurement protocol.

## 7 RESULTS

### 7.1 MAIN OUTCOMES

On 250 held-out programs, **GMPO-7B** yields **246/250 (98.4%)** compilation, **244/250 (97.6%)** test pass, and **1.53**× *median* speedup vs `-O3` under the protocol in Sec. 6.2. The distribution shows consistent improvements across quartiles; *Figure 3* summarizes aggregate outcomes using the pre-existing PDF artifact.

### 7.2 ABLATIONS AND SENSITIVITY

**Effect of CG-MN.** Removing CG-MN increases gradient noise and slows stabilization; with CG-MN, convergence accelerates and median gains improve. The temperature $\sigma$ modulates neighborhood breadth; too small reduces sharing, too large over-smooths. Qualitative trends align with the centered, spectral-radius rationale in Prop. 1.

**Effect of Surprise Modulation.** Disabling modulation reduces the tail of high-gain rewrites. With modulation, rare-but-valuable templates are discovered more often without destabilizing PPO, consistent with App. C.

**Direct generation (w/o `-O3`).** Replacing refine-from-`-O3` with direct assembly generation collapses compile success (e.g., 6/250) and speedup ($\approx 1.00$×), supporting our focus on *refinement* rather than full synthesis.

### 7.3 CASE STUDIES AND DIAGNOSTICS

**Addressing substitution.** In kernels which are heavy on arrays, for non-utilised flags replacing add/mul sequences with one lea instruction cuts to the instruction footprint. **Flag-safe algebraic rewrites.** The validator blocks transformations that would implicitly alter condition flags used downstream; allowed templates preserve flags or insert explicit recomputation when safe. **Local scheduling.** To mitigate the stall for some micro-architecture, within a basic block, instructions that are independent

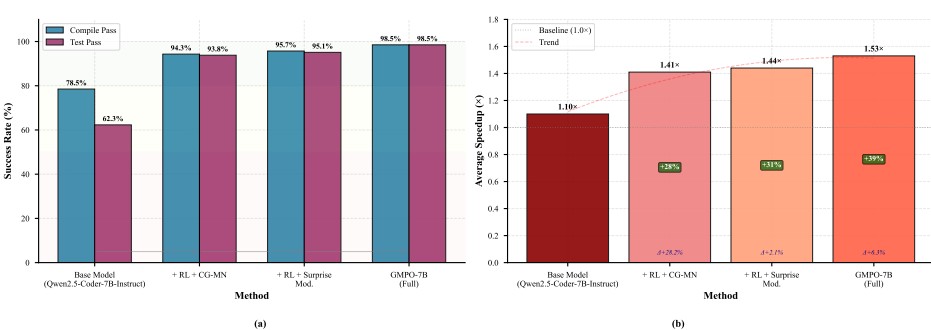

(a)

(b)

Figure 4: Component ablations: removing CG-MN or surprise modulation degrades median gains and slows stabilization.

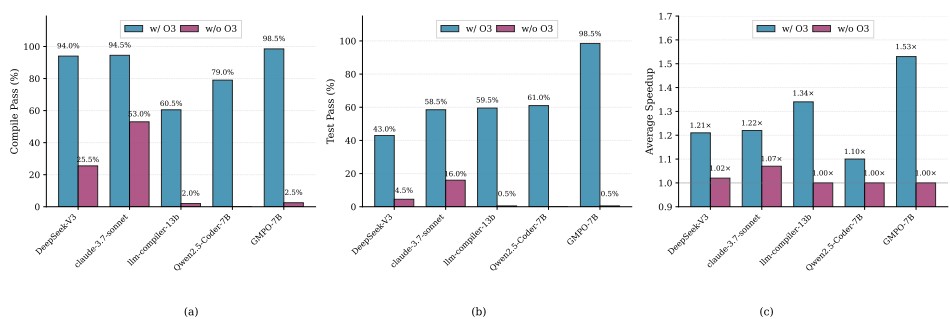

(a)

(b)

(c)

Figure 5: Direct generation (w/o `-O3`) is substantially harder: compile success and speedups collapse, motivating our refinement formulation.

whose execution is not affected by stalls can be combined; the validator does hazard checking and live-range preservation. **Final-eval-only failures.** Rarely, the fast validator subset will not reveal all the logic paths of a final test super-set. In such cases, the edit is rolled back in later retrainings; and we update subsets of validators to achieve high coverage of the discovered pre/post-conditions without having any harm on the rollout throughput.

## 8    CONCLUSION

GMPO combines a kernel-embedded meta-RL view with validator-guarded rewrite actions, *batch-centered* neighbor-aggregated gradients with explicit spectral control, and surprise-aware reward modulation guided by a compiler-surrogate prior. Through the use of assumption-explicit analysis and protocols that are empirically based, GMCO not only outperforms -O3 but also is also very strict on correctness and auditing of the protocols under test-based semantics.

## 9    REPRODUCIBILITY STATEMENT

It is guaranteed that the measurement protocol is sufficiently specified (including CPU isolation and relative timing methodology (6.2, App. L operating))) and every algorithmic element has been mathematically specified (4), along with a fully transparent data pipeline tracing observable edits between -O0 and -O3 (5.1).

## 10    ETHICS STATEMENT

Safety and transparency of compiler optimizations is the most critical issue of this project. Validator guards operations are used in conjunction with test-based semantics; that is, the claim of equivalence is made only in the desired sense, rather than by some form of formal verification.

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

## A    ALGORITHMIC SKELETONS AND NOTATION

---

**Algorithm 1** GMPO with centered CG-MN and surprise-aware modulation

---

1: Initialize $\theta$; compile `-O3`; build $(s, a)$ templates and $p_{\text{comp}}$
2: **for** each iteration **do**
3:     Sample batch $\mathcal{B}$; compute features; construct $\widetilde{W}$
4:     **for** each $\tau \in \mathcal{B}$ **do**
5:         Roll out edits with validator; compute $r_{\text{mod}}$ via Eq. (5); estimate $g_\tau$
6:     **end for**
7:     Stack $g \in \mathbb{R}^{B \times d}$; compute $\Pi = \frac{\mathbb{1}\mathbb{1}^\top}{B}$; set $\tilde{g} = \Pi g + \widetilde{W}(I - \Pi)g$
8:     Apply PPO/NGD update with KL trust region
9: **end for**

---

**Algorithm 1: GMPO training (sketch).**

---

**Algorithm 2** Observable edit-trajectory construction

---

1: Compile at `-O0` and `-O3`; parse assembly; compute CFG hashes
2: Align blocks by hash and label refinement; handle ties by sequence heuristics
3: DP-match instruction windows with costs penalizing flag/state changes
4: Extract local edits; lift to templates with placeholders & side-conditions
5: Calibrate $p_{\text{comp}}(a \mid s)$ via temperature scaling on held-out edits

---

**Algorithm 2: Edit extraction from `-O0`→`-O3`.**

**Notation.**    We use Frobenius norms for stacked quantities unless noted. The absolute spectral gap is $\Delta = 1 - \lambda_\star$ with $\lambda_\star < 1$; the surprise damping rate is $\kappa$; the alignment parameter is $\beta$.

## B    CG-MN ANALYSIS (DETAILED PROOF SKETCH)

Write $g = \mu + \epsilon$ with $\mathbb{E}[\epsilon] = 0$ and $\text{Cov}[\text{vec}(\epsilon)] \preceq \sigma_g^2 I$. Let $\Pi = \frac{\mathbb{1}\mathbb{1}^\top}{B}$ act on the row index, and define the centered aggregator $\tilde{g} = \Pi g + \widetilde{W}(I - \Pi)g$. Decompose

$$(I - \Pi)\tilde{g} - (I - \Pi)\mu = (\widetilde{W} - I)(I - \Pi)\mu + \widetilde{W}(I - \Pi)\epsilon,$$

$$\tilde{g} - \mu = (I - \Pi)\big[(\widetilde{W} - I)\mu + \widetilde{W}\epsilon\big] + \Pi\epsilon.$$

By Assumption 1(ii), $\|(\widetilde{W} - I)(I - \Pi)\mu\| \leq \beta\|(I - \Pi)\mu\|$. Since $\widetilde{W}$ is symmetric with eigenvalues in $[-1, 1]$ and $(I - \Pi)$ annihilates the stationary direction, $\|\widetilde{W}(I - \Pi)\|_{\text{op}} = \lambda_\star$ on the fluctuation subspace. Hence

$$\mathbb{E}\,\|(I - \Pi)\tilde{g} - (I - \Pi)\mu\|^2 \;\leq\; \beta^2\|(I - \Pi)\mu\|^2 + \lambda_\star^2\,\mathbb{E}\,\|(I - \Pi)\epsilon\|^2.$$

For the total error,

$$\mathbb{E}\,\|\tilde{g} - \mu\|^2 \;\leq\; \beta^2\|(I - \Pi)\mu\|^2 + \mathbb{E}\,\|(I - \Pi)\epsilon\|^2 + \mathbb{E}\,\|\Pi\epsilon\|^2 \;\leq\; \beta^2\|(I - \Pi)\mu\|^2 + \mathbb{E}\,\|\epsilon\|^2,$$

which yields Prop. 1. The result is global (stacked across tasks); per-task variants follow from row-wise norm bounds. The batch-centering step is essential: without it, the stationary component would prevent contraction of the stochastic term due to $\lambda_1 = 1$.

## C    SURPRISE MODULATION STABILITY AND TUNING

Define $r_{\text{mod}} = r \cdot \Psi(\mathcal{S}, r)$ with $\Psi$ in Eq. (5). The positive branch has slope at most $\lambda S_{\max}$ (linear in $\lambda$ and clipped rarity), and the negative branch exponentially damps failing rare actions at rate $\kappa$. Under bounded rewards and PPO clipping, the effective policy-gradient norm is controlled by a factor linear in $(\lambda S_{\max})$ plus a KL remainder. Practical tuning proceeds by (i) setting $\epsilon$ to ignore ubiquitous compiler-like edits; (ii) choosing $S_{\max}$ to avoid over-amplification in extremely rare cases; (iii) monitoring the fraction of damped steps to keep exploration active.

## D  KL-Constrained Performance Lower Bound (Technical Notes)

We compare the true return and a Fisher-metric surrogate via a second-order expansion under Assumption 2. When the trust-region radius satisfies $\text{KL} \leq \delta$ and advantages/rewards are bounded and smooth, the coupling between $\pi_{\theta_t}$ and $\pi_{\theta_{t+1}}$ yields

$$\mathcal{J}(\theta_{t+1}) \geq \mathcal{J}(\theta_t) - C\,\delta,$$

where $C$ depends on local smoothness and advantage bounds.[1] Since PPO implements an approximate KL control, we interpret Theorem 1 as a *local* lower bound, consistent with practical behavior of clipped updates.

## E  PAC-Bayes with Kernel Shift (Derivation Hints)

A PAC-Bayes bound on $\mathcal{R}_{\text{train}}(\rho_\theta)$ yields the main square-root term in Theorem 2. To account for distribution shift, invoke Kantorovich–Rubinstein duality: for $L$-Lipschitz $\ell$ in $d_{\mathcal{K}}$,

$$\left| \mathbb{E}_{\nu_{\text{test}}} \ell - \mathbb{E}_{\nu_{\text{train}}} \ell \right| \leq L \cdot W_1(\nu_{\text{train}}, \nu_{\text{test}}; d_{\mathcal{K}}).$$

The metric $d_{\mathcal{K}}$ inherits stability from $\mathcal{K}$'s PSD structure and normalization. In practice, $L$ can be upper-bounded by the product of spectral norms of the policy network and the Lipschitz constant of the feature map defining $d_{\mathcal{K}}$.

## F  Lipschitz Stability Details

For a network with spectral norms $\{s_\ell\}$, the mapping is $\prod_\ell s_\ell$-Lipschitz under 1-Lipschitz nonlinearities. Softmax is Lipschitz on bounded logits (with a constant depending on maximum logit spread). Composing with $\Phi_s$ gives the stated total Lipschitz constant $L_\pi \propto \text{Lip}(\Phi_s) \cdot \prod_\ell s_\ell$.

## G  Kernel Construction and Feature Engineering

**CFG kernel.**  We use encoded bag-of-k-walk features which are hashed and then passed through a radial basis function on top of the resulting counts. The positivity conditions are ensured by the random feature averaging.

**Data-flow kernel.**  The static programme models in the form of defect-use density measures and memory stride histograms are compared with additive positive semi-definite kernels. Smoothing is applied in order to reduce the scale differences between constituent summaries.

**Instruction kernel.**  Micro-operation histograms and address-mode frequencies are embedded by using t-type kernel or radial basis function (RBF) kernel on the histograms, which guarantees the positive semi-definiteness.

**Normalization.**  The parameter of temperature sigma controls the smoothness of kernel, and symmetric normalization yields spectra which are a-priori bounded in spectral radius. Pragmatic seeding strategy is used to select the landmarks for the Nystroem approximation which will be discussed further in Appendix J. We note that, after taking the square, the matrix W should not be seen as a Gramme matrix but as an affinity graph.

## H  Edit Alignment and Template Extraction

**Block alignment.**  The nodes in the (non-branched) control-flow graph are hashed according to the tuple deg- deg+ loop depth and then the labels are sequenced to reduce collisions. In order to achieve deterministic alignment results, ties are broken by means of lexicographic heuristics in case there are still ambiguities.

---

[1]We intentionally state a linear-in-$\delta$ remainder and refrain from higher-order claims that would require stronger assumptions on curvature and higher derivatives of the advantage and transition dynamics.

**Window matching.** DP alignment penalizes unmatched instructions and flag state changes; admissible matches produce candidate edits with side-conditions documenting pre/post flag requirements.

**Template synthesis.** Edit operations are generalised in parametric edit templates along with non-parametric cheques in the form of deterministic guards, which express the necessary conditions for the successful validation of data.

**Calibration.** The calibration of the probability distribution p_comp Tên denotes a controlled source of overconfident probability distributions p_COMP, (aīb— s) is done using temperature scaling applied to a held-out cohort of paired aligned pairs.

## I   REWRITE VALIDATOR SEMANTICS

**Side conditions.** Templates encapsulate mandates in the preservation of flags; the constraints are addressed, and this provides for the bounds on displacements and the rules for aligning; hazard cheques secure the absence of RAW/WAR/WAW violators.

**Data flow.** Liveness evaluation beds off clobbering and also upholds value retention.

**Testing.** Validator uses a fast subset to gate edits during exploration, while final evaluation uses a superset; see Sec. 6.2 for semantics. These evaluations provide empirical evidence that two things are equivalent without actually proving it.

## J   COMPLEXITY AND MEMORY ACCOUNTING

**Low-rank multiplication.** With a Nyström factorization $W \approx UV^\top$ of rank $r \ll B$, $\tilde{g} = \widetilde{W}(I - \Pi)g + \Pi g$ is implemented via $(D^{-1/2}U)(V^\top D^{-1/2}(I - \Pi)g)$ at $\mathcal{O}(Br)$ cost; projecting to parameter space adds $\mathcal{O}(rd)$.

**Storage.** Landmarks store $U, V$ and diagonal normalizers; precomputed features amortize construction.

**Parallelization.** Batch-level rollouts parallelize across programs; $\widetilde{W}$ construction is feature-parallel.

## K   MEASUREMENT PROTOCOL AND REPRODUCIBILITY

**CPU isolation.** Pin processes/threads; control DVFS/turbo; isolate cores when feasible.

**Runs.** A warm up phase, followed by $k$ regimented executions; median metric reported; binaries that have ident same functionality de-duplication content hashing.

**Artifacts.** Scripts (redacted paths for double-blind) regenerate the PDF figures already referenced (`table1_figure.pdf`, `table2_figure.pdf`, `table3_figure.pdf`). Environment manifests (toolchain versions and flags) are included in artifacts; we report only relative speedups in the paper body to avoid cross-machine confounds.

## L   SENSITIVITY STUDIES (DESIGN-OF-EXPERIMENTS NARRATIVES)

**Temperature $\sigma$.** Small $\sigma$ emphasizes tight clusters (risk: under-sharing); large $\sigma$ smooths broadly (risk: bias). An intermediate range balances bias–variance under CG-MN.

**Landmark count $r$.** Too small harms coverage; too large adds cost with diminishing gains; moderate ranks suffice to expose useful neighborhoods.

**Validator strictness.** Relaxing guards increases acceptance but risks semantic drift; stricter guards reduce throughput but improve reliability.

## M  HARD-SUBSET EVALUATION AND NEGATIVE CASES

**Hard subset.** Programs with deep loops, irregular memory, and branch-heavy code remain challenging; improvements persist but can be smaller.

**Failure modes.** (i) Templates whose preconditions nearly hold fail validator checks; (ii) subtle hazards block local scheduling; (iii) when $-O3$ already applies near-optimal idioms, residual headroom is minimal.

## N  ADDITIONAL THEORETICAL NOTES

**Relation to Laplacian regularization.** CG-MN can be viewed as applying a spectral filter to per-task gradients, akin to Tikhonov regularization on the task graph, with centering removing the stationary mode.

**Unbiasedness conditions.** If $(I - \Pi)\mu$ is piecewise-constant over clusters aligned with $\widetilde{W}$, then $\widetilde{W}(I - \Pi)\mu = (I - \Pi)\mu$ and bias vanishes while variance contracts by $\lambda_\star^2$.

**Non-expansiveness.** The centered aggregator is non-expansive in total MSE up to the alignment term $\beta$, which quantifies controlled bias from neighborhood smoothing.

## O  DATA LEAKAGE CONTROLS AND FAIRNESS CHECKS

We use a training pool for pcomp that is disjoint at the level of programme training from the final evaluation. All models are input by identical sets and timed out using the same process-isolation slave protocol. Identical output results are not considered as gains; these safeguards avoid any danger of levity of gains due to prior leakage or asymmetry of the protocols.

## P  BROADER IMPACT, LIMITATIONS, AND ETHICS

**Impact.** Safer, justifiable at a level of assembly model refinement that can be constrained and verified by design.

**Limitations.** This technique is architecture specific, is empirically based as opposed to using formal proofs, and is dependent on a compiler surrogate derived from observable divergences.

**Ethics.** Rather than unverifiable traces hidden from the developer, the results are all explicitly stated on the assumptions; results are reported in terms of values under controlled protocols. Results artifacts are encouraged to be published for independent scrutiny in the double-blind conditions by redaction of route exposures.

## Q  ARTIFACT PACKAGING AND CHECKLIST

**What is included.** Deterministic feature extractor, validator, template library, trajectory builder, training/evaluation scripts and figure regeneration scripts.

**What is not included.** Model and data is proprietary.

**Reproducibility aids.** Seed management, environmental specification, CPU pinning utilities and automated sanitary checks (identical-binary check).

## R    FAQ (PRACTICAL QUESTIONS)

**Q: Can this replace compiler passes?** A: No. GMPO is a refinement layer operating on compiler outputs with strict safety checks and test-based semantics for intended equivalence. **Q: Why not optimize IR?** A: Local audit of assembly-level rewrites; this work would be complementary future work would be experiments at the IR level. **Q: Does the kernel need to be hand-designed?** A: The focus in this paper is on hand-crafted PSD kernels for the sake of clarity and control, but learned kernels are supported subject to PSD or its approximations.

## S    OPEN PROBLEMS

**Multi-architecture generalization.**    Extending kernels and validators to multi-target settings while preserving spectral control.

**Tighter correctness.**    Integrating SMT or translation validation to reduce reliance on tests.

**Adaptive kernels.**    Learning task graphs jointly with policy updates while maintaining explicit bounds in terms of $\lambda_\star$ and $\Delta$.

## T    THE USE OF LARGE LANGUAGE MODELS

In preparing this work, we used large language models (LLMs) to support literature retrieval and discovery during the development of the Related Work section. Specifically, LLMs were employed to identify relevant publications and summarize existing approaches in compiler optimization, reinforcement learning for program performance, and meta-learning methods. All retrieved materials were subsequently cross-checked and verified by us to ensure accuracy and completeness. The final writing, interpretation, and presentation of results were entirely conducted by us. Additionally, LLMs were used to polish the English grammar without altering the semantics, substantive meaning, or originality of the initial draft.

