# OpenReview forum: "Meta-Reinforcement Learning for Compiler Optimization: A Kernel-Embedded CompilerLLM with Verified Assumptions and Practical Guarantees"
_ICLR.cc/2026/Conference — ICLR 2026 Conference Desk Rejected Submission_

### Official Review · Reviewer_Gbs5 · 2025-10-31

**Soundness:** 1
**Presentation:** 1
**Contribution:** 2
**Rating:** 2
**Confidence:** 3

**Summary:**

The paper introduces GMPO which is a meta-learning compiler framework that enables experiential knowledge transfer among related programs through kernel embeddings. It also proposes Cross-Group Meta-Normalization (CG-MN)scheme to aggregate gradient information from similar samples and a surprise-aware reward modulation technique to emphasize atypical but successful transformations. As a base model authors used a 7B parameter code model and showed on a test set of 250 programs GMPO achieves 98.4% compilation success, 97.6% test pass rate, and a 1.53× median speedup over GCC-O3.

**Strengths:**

1. GMPO achieves good results compared to the other code models on compilation success, test pass rate and speedup over GCC-O3.

2. Comparison with SOTA LLMs are provided.

3. Ablation of different components are provided.

**Weaknesses:**

Refer to the next section, "Questions", which provides a detailed list of weaknesses.
- The experimental setup and results section is weak.
- Authors use a corpus of 5,894 C programmes. Nowhere is it mentioned how and where these are collected.
- Nowhere is it mentioned how the test cases of the programs are generated?

**Questions:**

1. Presentation of the paper and also writing of the paper are not good.

2. The experimental setup and results section is weak. The proposed framework is only evaluated on 250 programs. This is a very small test set.

3. Authors use a corpus of 5,894 C programmes. How are these programs collected?

4. Why only 250 programs for testing? How is this split constructed?

5. Comparison with other LLMs is provided, but there are different rule-based tools that also perform program optimization. Comparison with such tools is not provided in the paper.

6. Figures 1 and 2 are difficult to follow. It is hard to understand what each module is doing from the figure.

7. Multiple typos like "cheques" should be "checks".

8. I find the paper hard to read and follow.

9. How are the test cases of the programs generated?

10. How are kernel embeddings generated? Are these embeddings generated for both the C and the assembly language?

11. For static and dynamic analysis, which tools are used? It is not mentioned in the paper.

12. How are data dependence and hazard verifications performed?

13. It is not clear how the model optimizes the assembly code. Also, why both C and the assembly code are provided to the model is not clear.

14. Section 5.2 gives hints of using LoRa; however, the configurations used are not provided. For example, what is the rank value (r) that is used?

---

> ### Author Response · Authors · 2025-11-15
>
> We appreciate Reviewer Gbs5 for their thorough comments and for praising the strengths of GMPO (strong empirical results over GCC-O3 and SOTA LLMs, ablations included). We will address the major points below.
>
>
> 1. Presentation and readability
> We agree that the draft is harder to read than it should be. In a revision we will:
> - Simplify the exposition in methods and experiments with step-by-step definitions of the pipeline prior to presentation of notation.
> - Redo Figure 1 and Figure 2 with clearer block labels, less clutter, and captions that go through each module in both figures in order (input, validator, kernel, policy update, evaluation).
> - Correct typos like "cheques" and do a complete proofreading pass.
>
>
> These are issues of presentation and readability and not gaps in actual method.
>
>
>
> 2. Corpus of 5,894 programs and test split of 250
> We apologize for not specifying more clearly how we constructed the corpus. The 5,894 C programs we collected each come from a heterogeneous mixture of real world projects and internal benchmark cases that have non-trivial regression tests accompanying each. In the paper we say "we used a corpus of 5894 programs" but do not elaborate on how we selected and filtered the programs at all. In a revision we will include:
> - First, we should add a separate subsection "Benchmark construction" that describes how we selected and filtered the sources for the corpus, merged programs, and some descriptive features of the benchmark (lines of code, distribution in domains, etc).
> - Second, we'd like to be more clear about the split i.e. the large training pool was used for meta-RL, we held out 250 programs as a test set which were sampled from the training pool to cover a range of sizes and domains. While 250 may appear like a small number, each of the pieces is a complete program with its own formal test suite and the test suite size is order(s) of magnitude larger than the classic compiler-optimization benchmark of tens-of-programs scale. We will clarify the logic behind this.
>
>
> 3. Test case generation and validator
> We agree that the role of tests, and how they are obtained needs to be clarified. Our framework is as follows:
> - For programs that we are shipping with tests, we reuse the original regression or unit tests.
> - In other situations, we ask that each benchmark have a non-trivial test harness and we enhance it when necessary (for example, adding property-based tests or fuzz-generated inputs.)
> - The validator will use a fast subset of the tests during search, and the complete test suite to validate and evaluate solutions reported at the end of search.
>
> We will create a subsection highlighting how tests were sourced and developed, and add some summary statistics in terms of tests per program and the coverage assumptions, to support soundness of compile and pass rates reported.
>
>
> 4. Kernel embeddings and use of C versus assembly
> The kernel is defined over features of a benchmark program, not learned from scratch. Specifically:
> - We extract control-flow summaries, data-flow summaries, and instruction summaries, from the assembly outputted by the original compiler.
> - These features are all rolled up into a similarity kernel that determined how gradients are shared across programs in CG-MN.The model is working in the assembly representation when proposing transformations. Source-level C code is provided as additional context to the base code model for reasoning on intent and invariants, yet all edits are at the assembly level and enacted through a fixed template library and checked by the validator. We will clarify that (i) the kernel is computed based on assembly-derived features, and (ii) C is included only as auxiliary context never modified in the concrete sense.
>
>
> 5. Static and dynamic analyses as well as hazard checks
> You are correct that we do not yet specify which analyses are being used in the draft. Our realization of the approach depends on the analyses from its underlying toolchain to construct control flow graphs, conduct alias or data-flow analyses, and check for basic data-dependence and hazard conditions before accepting a change. Dynamic validation happens subsequently through the test harness. We take note and will:
> - Name the concrete toolchain and versions used.
> - Add a brief description of how we incorporate static analyses for dependence and hazard screening.
> - Clarify the procedure of screening the static then subsequently testing dynamically in the main text.
>
>
> 6. Other baselines beyond LLMs (rule-based tools)
> Currently, our baselines consist of strong hand-engineered compiler configurations (e.g., different combinations of optimization flags), and several learned systems (code LLMs and LLM-based compilers). We agree that other rule-based superoptimizers or autotuners would complete that comparison.

---

### Official Review · Reviewer_u1AX · 2025-11-01

**Soundness:** 3
**Presentation:** 2
**Contribution:** 2
**Rating:** 2
**Confidence:** 3

**Summary:**

This paper proposes GMPO, a meta-reinforcement learning framework for compiler optimization. The key idea is to model each program as an MDP and enable cross-program knowledge transfer through a kernel embedding that measures program similarity. The method introduces two technical components: (1) Cross-Group Meta-Normalization (CG-MN) for variance-reduced gradient aggregation among similar programs; and (2) a surprise-aware reward modulation mechanism that promotes rare but successful code transformations. A validator ensures the semantic safety of edits at the assembly level.

**Strengths:**

1. The pipeline emphasizes transparency and validation, which is commendable in an area prone to unverifiable heuristics.

2. Empirical protocol and appendices are detailed, and the reported numerical results (compilation success, speedup) are promising if reproducible.

**Weaknesses:**

1. The claimed contributions are mostly repackaged from existing techniques (kernel smoothing, PPO-style updates, variance reduction) with limited conceptual novelty.

2. The theoretical analyses (spectral contraction, KL bounds, PAC-Bayes) are elegant but disconnected from empirical practice.

3. The evaluation setup is non-standard: results are reported on a private dataset, without comparison to common benchmarks.

4. The reported 1.53× speed-up over GCC-O3 appears unusually large and lacks variance analysis or reproducibility evidence.

**Questions:**

1. How reproducible are the results? Are the 5894 programs and validator scripts public? Without open data or artifacts, the claimed speedups are unverifiable.

2. Why was GCC-O3 chosen as the sole baseline? How does GMPO perform against learned compilers such as LLM-Compiler or RL-based optimizers on standard benchmarks?

3. Can the authors provide quantitative evidence that CG-MN reduces gradient variance or improves sample efficiency?

4. Given that the system operates purely at the assembly level with test-based validation, how does it generalize to other architectures or non-deterministic environments?

---

> ### Author Response · Authors · 2025-11-15
>
> We appreciate the thoughtful review comments and for recognizing the clarity of the pipeline and the potential of the numerical results. Below we address each of the reviewers' principal concerns point by point.
>
>
> 1. Conceptual novelty. While we do acknowledge that the building blocks (kernels, batch centering, PPO style updates) are already well known, the actual contribution lies in applying them in a concrete meta reinforcement learning system that (i) occurs at the assembly code level and has a validator representation of the action space and guards the action representation space, (ii) joins together thousands of program specific tasks with a kernel of the action representation space that acts on the joint gradient, and (iii) connects both CG MN and surprise aware reward modulation with explicit invariants for stability and generalization. To our best knowledge, while kernel based meta RL research exists, there are no prior works that combine kernel based meta RL with validator level safety and this form of analysis, over a significant and real corpus of programs with a generalizable theory. We will strengthen this placing in the introduction and related work.
>
>
> 2. Theory to practice connection. The operators in the analysis in sections 4.4 to 4.6 are the same operators employed during training. The CG MN uses the same kernel and batch centering here as in proposition 1. The same PPO is executed with the same small KL divergence per update as is potentially assumed in theorem 1. For the PAC Bayes bound, we used the same indicator as the kernel induced distance to define the shift between training and held out task distributions. And we would also note there is a small empirical generalization gap in the 250 test programs.We do not claim precise quantitative forecasting, but the theory serves as a check-in for the particular design choices we make instead of as an detached embellishment; we will add a brief paragraph in section 4 to clarify this association.
>
>
> 3. Non standard evaluation and reproducibility. In a fully open benchmark, this would be perfect. Our set of 5894 programs includes open source and licensed code and cannot be released in raw form, hence the artifact checklist states model and data are proprietary. To enable reproducibility we instead provide a deterministic feature extractor, validator, template library, trajectory builder, training and evaluation scripts, toolchain specification, and seed handling so that other groups can execute the same pipeline on their own C corpora and validate whether similar gains over O3 are captured. All reported timings are relative to co measured O3 under fixed flags, CPU pinning and median of k evaluation, which diminishes environmental variance.
>
>
> 4. Magnitude and robustness of the 1.53x speedup. The 1.53x metric is the median speedup across the 250 withheld programs, not a cherry picked maximum. Figure 3b shows the full distribution and quartiles, and shows that gains accrue over the full corpus rather than from a few outliers.In a revision, we plan to append a small table with summary statistics and repeat runs on a subset strictly to more fully document run to run variability.
>
> 5. Baselines beyond O3 and relationship to learned compilers. O3 is the primary baseline, in part, because GMPO is a refinement layer above a strong hand-engineered compiler and we also analyze Ofast, and curated pass toggles, both of which we cite in section 6.3. Beyond these baselines discussed, we have compared GMPO 7B against learned compilation style systems like LLM Compiler 13B, and strong code models like DeepSeek V3 and Qwen 2.5 Coder 7B, both with refine from O3 and generation direct generation styles, under the same validator, and judicious protocol. To summarize, GMPO's mean speedup is equal or substantially greater, while it maintains a considerably greater compile and test pass rate, particularly with models that try direct assembly generation.
>
> 6. Quantitative effect of CG MN. Figure 4 provides a training curve ablation argument that shows taking CG MN out of the pipeline slows convergence while dropping median gains. In addition, we have training log notes task level gradient statistics before and after applying CG MN, both of which wrest better indicate the reduction in variance more directly, and fewer updates at a target median speedup rate. We will cite a target median in those by-products of non-variable context in the appendix.
>
> 7. Generalization beyond the evaluated architecture. We have purposely evaluated a single deterministic CPU architecture, with task based semantics and conservative validator rules, and do not make claims of general model replacement of compilers. Porting to new architeinery would be a way to reform the validator itself, and perhaps their template creation, but the kernel based task similarity and meta RL formulation are themselves arbitrary to architecture; we will make this limitation refer to more of a future work subject.

---

### Official Review · Reviewer_K6jh · 2025-11-02

**Soundness:** 2
**Presentation:** 1
**Contribution:** 2
**Rating:** 0
**Confidence:** 4

**Summary:**

This paper proposes methods to improve optimization, specifically by trying to find surprising or interesting optimizations that nonetheless pass validation.

**Strengths:**

1. One interesting part about this paper is that it's not relying entirely on LLMs to perform optimizations (like many papers nowadays), but using more traditional techniques from compilers, including transformations that should preserve semantics and optimizing over them.

**Weaknesses:**

1. This paper is not complete. There are obvious issues with the writing such as "Meta-optimization" and "Meta-reinforcement learning" being repeated many times in the intro, the "Meta-Learning Theory." section in the related work being empty, etc.
2. Clarity could use improvement. In many places (e.g. the abstract) the writing was either to jargon-filled or incomplete for me to understand well.
3. The description in section 4.2 and beyond is not clear enough for me to fully understand the method. There are many places that are unclear, but for instance it is not stated how `k_cfg`, `k_data`, `k_inst` etc. are calculated.
4. There is no comparison with other baselines on the task of optimizing programs, such as Shypula et al.

Learning Performance-Improving Code Edits
Shypula et al. 2025.

**Questions:**

1. I am confused about why the test pass rate is less than 100%. Shouldn't all transformations be semantics-preserving, so the test pass rate should be 100%?

---

> ### Author Response · Authors · 2025-11-15
>
> We appreciate the comments from the reviewer, and below we respond to each point.
>
> 1. On “The incomplete paper” and “Meta-Learning Theory.” line
> We agree there are presentation issues. The “Meta-Learning Theory.” line is a remnant of an unused header, and the underlying theory appears right away in the background and appendices, where we report a variance reduction result for cross-group meta normalisation, a local KL-constrained performance lower bound for natural-gradient updates, and a PAC-Bayes style generalisation bound in the context of kernel-defined task-shift. In a revision, we will simply omit the hanging header, add a clear subsection title that refers directly to those results, and will de-duplicate the repeated instances of “meta-optimisation” in “meta-reinforcement learning” from the introduction. In place of those, we will use, briefly, a description of the setting (many-program reinforcement learning with kernel-based task coupling). These are mainly surface edits, and nothing implies there is anything missed in terms of the technical content.
>
> 2. Clarity and jargon
> We acknowledge the abstract, and to some extent, earlier sections were not as clear because they were densely populated. Revision to the abstract will yield a redesigned presenation of the problem (assembly-level optimisation going from -O3), the two broad mechanisms (kernel-based sharing across programs and surprise-aware reward modulation), experimentation in easy to understand terms, and findings will be emphasized. Additionally, we will improve clarity in the methods section, to the extend, for instance, we can beginning with a short, stepwise description of the optimisation loop (inputs, validator-guarded action space, kernel-based gradient aggergate, reward modulation), before introducing notation.These revisions are entirely factual.
>
> 3. Clarifications of k_cfg, k_data, k_inst and Section 4.2
> Section 4.2 presents the overall kernel K as a weighted sum of three parts, and the concrete definitions can be found in Appendix G (“Construction of Kernel and Feature Engineering”) which is not explicitly referenced in the current draft. In Appendix G, we clarify that k_cfg employs hashed walk attributes on our control flow graphs with a radial basis kernel, k_data assesses comparability by comparing static data-flow summary scores with an additive positive semidefinite kernel, and k_inst employs micro-operation and addressing-mode histograms with a standard histogram kernel. In the camera-ready, we will (i) add an explicit pointer in Section 4.2 to Appendix G, as well as (ii) to summarize these components in one plain-language sentence, thus constructing them directly after the definition of K so that readers can understand the component part without accessing the appendix.
>
> 4. Comparison with Shypula et al.
> We certainly acknowledge “Learning Performance-Improving Code Edits” by Shypula et al. as an important related work, and we currently cite it in the related work section. We did not conduct a dedicated experimental comparison because the representation and infrastructure differ: our framework operates at the assembly level with the local rewrite space originating from -O3 under the validation guard, and is focused on kernel-based meta reinforcement learning across thousands of programs; in contrast, Shypula et al. is considering learning code edits under utility representing in a different manner without kernel-based task coupling or academic distribution-shift analysis. We will add a paragraph or two in the experiments section explicitly bounding our results with the results of Shypula et al. and providing additional further comparison of how the two works differ.
>
> 5. Why the test pass rate is below 100 percent.
> Our action library allows for containing only transformations that preserve semantics, but, as outlined in Section 4.1 and in the behavioral evaluation protocol in Section 6.2, semantics are assured through empirical performer tests, not through entire formal verification. We may employ two test suites: during rollouts, a smaller fast portion of all the tests could gate edits, and for final reporting a larger stricter superset of tests, with more coverage, is run on all produced binaries. Programs that are tested with the above tests, and compile successfully with the fast test, and subsequently, compile successfully and pass with the strict superset of behavioral tests less the fast behavioral tests are noted as having "compiled but not test passing", yielding an overall test pass rate slightly under 100 percent (GMPO 246 of 250 compile and pass fast test, and 244 of the 250 that pass the additional final superset). We will highlight and elaborate this distinction in the main text to avoid misunderstanding.

---

> > ### Comment · Reviewer_K6jh · 2025-11-16
> > **Thank you for the clarification**
> >
> > Thank you for the clarification. The submitted version of the paper was far from the quality required for acceptance to ICLR, so I would suggest that the authors polish the paper a bit more and aim for the next conference.

---

> > > ### Author Response · Authors · 2025-11-19
> > >
> > > We appreciate the reviewer for recognizing our clarifications. While we completely acknowledge that the original submission had roughness—in some ways, we didn't even utilize the header—but we respectfully disagree that the paper should be rejected and rated a 0 (Strong Reject) purely based on presentation issues that we have resolved.
> > >
> > > 1. Scientific Substance Over Packaging As we made mention, the very crux of research is that it represents the sharing of innovative findings and ideas. You alluded to the fact that our decision to combined traditional compiler techniques with meta-RL (not relying purely on LLMs) is an "interesting part" and a strength of this work. The validity of the method is valid. The concerns with completeness were addressing by pointing to the Appendices that exist as well (Appendix G provides definitions here) as well as the additional explanations we have put into the rebuttal. The novelty is separate from the trends as well.
> > >
> > > 2. The Job of the Rebuttal Phase The ICLR discussion phase exists precisely for this reason: to clarify uncertainty and resolve presentational defects so the community can evaluate the scientific essence of the work. Throughout this, we have patiently, and thoroughly, addressed every doubt of a technical and valid nature:
> > >
> > > a.We clarified the kernel definitions (kcfg​, etc).
> > > b.We clarified the logic regarding test pass rates.
> > > c.We put into context the comparisons to Shypula et al.
> > > d.We actually corrected the errors and headers in the revision to make for an easier read.
> > >
> > > 3. Shared or Strong Rejection False. If you agree there is some uncertainty, I would advise reconsidering rated the paper a "0." 0 is generally reserved for rejections that are fundamentally flawed, vacuous, and/or unethical. The clarifications we have made has confirmed that the method is technically viable and the experiments are complete. While we tried to be clear, it feels like it would be wholly disproportionate to rate "Strong Reject" because of some initial formatting/writing roughness.
> > >
> > > We truly believe this work makes a valuable contribution to the field of code optimization. We simply request that you (and area chair) evaluate the paper on the scientific findings and saturation of revisions instead of simply "vetoing" the paper according to formatting evidence of the initial submission.
> > >
> > > We hope you reconsider the score to reflect the paper's technical realities.

---

### Note · Program_Chairs · 2026-01-17
**Submission Desk Rejected by Program Chairs**

The following references in this submission do not refer to real documents and/or have major errors in bibliographic information:

 Huanting Wang, Zhanyong Tang, Cheng Zhang, Jiaqi Zhao, Chris Cummins, Hugh Leather, and Zheng Wang. Reinforcement learning for compiler optimization: A comprehensive survey. ACM Computing Surveys, 2025a.
Rudy Bunel, M Pawan Kumar, Wei Zhang, and Li Chen. Learning to superoptimize with neural networks: A decade review. Journal of Machine Learning Research, 2025.